# ORDER LEARNING AND ITS APPLICATION TO AGE ESTIMATION

**Kyungsun Lim, Nyeong-Ho Shin, Young-Yoon Lee, and Chang-Su Kim**
School of Electrical Engineering, Korea University and Samsung Electronics Co., Ltd
{kslim, nhshin, cskim}@mcl.korea.ac.kr, yy77lee@gmail.com

## ABSTRACT

We propose order learning to determine the order graph of classes, representing ranks or priorities, and classify an object instance into one of the classes. To this end, we design a pairwise comparator to categorize the relationship between two instances into one of three cases: one instance is 'greater than,' 'similar to,' or 'smaller than' the other. Then, by comparing an input instance with reference instances and maximizing the consistency among the comparison results, the class of the input can be estimated reliably. We apply order learning to develop a facial age estimator, which provides the state-of-the-art performance. Moreover, the performance is further improved when the order graph is divided into disjoint chains using gender and ethnic group information or even in an unsupervised manner.

## 1 INTRODUCTION

To measure the quality of something, we often compare it with other things of a similar kind. Before assigning 4 stars to a film, a critic would have thought, "It is better than 3-star films but worse than 5-stars." This ranking through pairwise comparisons is done in various decision processes (Saaty, 1977). It is easier to tell the nearer one between two objects in a picture than to estimate the distance of each object directly (Chen et al., 2016; Lee & Kim, 2019a). Also, it is easy to tell a higher pitch between two notes, but absolute pitch is a rare ability (Bachem, 1955).

Ranking through comparisons has been investigated for machine learning. In learning to rank (LTR), the pairwise approach learns, between two documents, which one is more relevant to a query (Liu, 2009). Also, in ordinal regression (Frank & Hall, 2001; Li & Lin, 2007), to predict the rank of an object, binary classifications are performed to tell whether the rank is higher than a series of thresholds or not. In this paper, we propose order learning to learn ordering relationship between objects. Thus, order learning is related to LTR and ordinal regression. However, whereas LTR and ordinal regression assume that ranks form a total order (Hrbacek & Jech, 1984), order learning can be used for a partial order as well. Order learning is also related to metric learning (Xing et al., 2003). While metric learning is about whether an object is 'similar to or dissimilar from' another object, order learning is about 'greater than or smaller than.' Section 2 reviews this related work.

In order learning, a set of classes, $\Theta = \{\theta_1, \theta_2, \cdots, \theta_n\}$, is ordered, where each class $\theta_i$ represents one or more object instances. Between two classes $\theta_i$ and $\theta_j$, there are three possibilities: $\theta_i > \theta_j$ or $\theta_i < \theta_j$ or neither (*i.e.* incomparable). These relationships are represented by the order graph. The goal of order learning is to determine the order graph and then classify an instance into one of the classes in $\Theta$. To achieve this, we develop a pairwise comparator that determines ordering relationship between two instances $x$ and $y$ into one of three categories: $x$ is 'greater than,' 'similar to,' or 'smaller than' $y$. Then, we use the comparator to measure an input instance against multiple reference instances in known classes. Finally, we estimate the class of the input to maximize the consistency among the comparison results. It is noted that the parameter optimization of the pairwise comparator, the selection of the references, and the discovery of the order graph are jointly performed to minimize a common loss function. Section 3 proposes this order learning.

We apply order learning to facial age estimation. Order learning matches age estimation well, since it is easier to tell a younger one between two people than to estimate each person's age directly (Chang et al., 2010; Zhang et al., 2017a). Even when we assume that age classes are linearly ordered, the proposed age estimator performs well. The performance is further improved, when classes are

divided into disjoint chains in a supervised manner using gender and ethnic group information or even in an unsupervised manner. Section 4 describes this age estimator and discusses its results. Finally, Section 5 concludes this work.

## 2  RELATED WORK

**Pairwise comparison:** It is a fundamental problem to estimate the priorities (or ranks) of objects through pairwise comparison. In the classic paper, Saaty (1977) noted that, even when direct estimates of certain quantities are unavailable, rough ratios between them are easily obtained in many cases. Thus, he proposed the scaling method to reconstruct absolute priorities using only relative priorities. The scaling method was applied to monocular depth estimation (Lee & Kim, 2019a) and aesthetic assessment (Lee & Kim, 2019b). Ranking from a pairwise comparison matrix has been studied to handle cases, in which the matrix is huge or some elements are noisy (Braverman & Mossel, 2008; Jamieson & Nowak, 2011; Negahban et al., 2012; Wauthier et al., 2013). On the other hand, the pairwise approach to LTR learns, between two documents, which one is more relevant to a query (Liu, 2009; Herbrich et al., 1999; Burges et al., 2005; Tsai et al., 2007). The proposed order learning is related to LTR, since it also predicts the order between objects. But, while LTR sorts multiple objects with unknown ranks and focuses on the sorting quality, order learning compares a single object $x$ with optimally selected references with known ranks to estimate the rank of $x$.

**Ordinal regression:** Ordinal regression predicts an ordinal variable (or rank) of an instance. Suppose that a 20-year-old is misclassified as a 50-year old and a 25-year old, respectively. The former error should be more penalized than the latter. Ordinal regression exploits this characteristic in the design of a classifier or a regressor. In Frank & Hall (2001) and Li & Lin (2007), a conversion scheme was proposed to transform an ordinal regression problem into multiple binary classification problems. Ordinal regression based on this conversion scheme has been used in various applications, including age estimation (Chang et al., 2010; 2011; Niu et al., 2016; Chen et al., 2017) and monocular depth estimation (Fu et al., 2018). Note that order learning is different from ordinal regression. Order learning performs pairwise comparison between objects, instead of directly estimating the rank of each object. In age estimation, ordinal regression based on the conversion scheme is concerned with the problem, "Is a person's age bigger than a threshold $\theta$?" for each $\theta$. In contrast, order learning concerns "Between two people, who is older?" Conceptually, order learning is easier. Technically, if there are $N$ ranks, the conversion scheme requires $N-1$ binary classifiers, but order learning needs only a single ternary classifier. Moreover, whereas ordinal regression assumes that ranks form a total order, order learning can be used even in the case of a partial order (Hrbacek & Jech, 1984).

**Metric learning:** A distance metric can be learned from examples of similar pairs of points and those of dissimilar pairs (Xing et al., 2003). The similarity depends on an application and is implicitly defined by user-provided examples. If a learned metric generalizes well to unseen data, it can be used to enforce the desired similarity criterion in clustering (Xing et al., 2003), classification (Weinberger et al., 2006), or information retrieval (McFee & Lanckriet, 2010). Both metric learning and order learning learn important binary relations in mathematics: metric and order (Hrbacek & Jech, 1984). However, a metric decides whether an object $x$ is similar to or dissimilar from another object $y$, whereas an order tells whether $x$ is greater than or smaller than $y$. Thus, a learned metric is useful for grouping similar data, whereas a learned order is suitable for processing ordered data.

**Age estimation:** Human ages can be estimated from facial appearance (Kwon & da Vitoria Lobo, 1994). Geng et al. (2007) proposed the aging pattern subspace, and Guo et al. (2009) introduced biologically inspired features to age estimation. Recently, deep learning has been adopted for age estimation. Niu et al. (2016) proposed OR-CNN for age estimation, which is an ordinal regressor using the conversion scheme. Chen et al. (2017) proposed Ranking-CNN, which is another ordinal regressor. While OR-CNN uses a common feature for multiple binary classifiers, Ranking-CNN employs a separate CNN to extract a feature for each binary classifier. Tan et al. (2018) grouped adjacent ages via the group-n encoding, determined whether a face belongs to each group, and combined the results to predict the age. Pan et al. (2018) proposed the mean-variance loss to train a CNN classifier for age estimation. Shen et al. (2018) proposed the deep regression forests for age estimation. Zhang et al. (2019) developed a compact age estimator using the two-points representation. Also, Li et al. (2019) proposed a continuity-aware probabilistic network for age estimation.

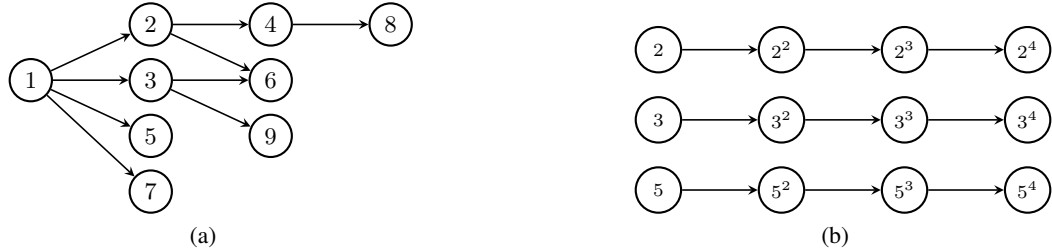

(a)                                                                                    (b)

Figure 1: Examples of order graphs, in which node $n$ precedes node $m$ ($n \to m$), if $n$ divides $m$. For clarity, self-loops for reflexivity and edges deducible from transitivity are omitted from the graphs.

## 3    ORDER LEARNING

### 3.1    WHAT IS ORDER?

Let us first review mathematical definitions and concepts related to order. An *order* (Hrbacek & Jech, 1984; Bartle, 1976), often denoted by $\leq$, is a binary relation on a set $\Theta = \{\theta_1, \theta_2, \cdots, \theta_n\}$ that satisfies the three properties of

- Reflexivity: $\theta_i \leq \theta_i$ for every $\theta_i \in \Theta$;
- Antisymmetry: If $\theta_i \leq \theta_j$ and $\theta_j \leq \theta_i$, then $\theta_i = \theta_j$;
- Transitivity: If $\theta_i \leq \theta_j$ and $\theta_j \leq \theta_k$, then $\theta_i \leq \theta_k$.

In real-world problems, an order describes ranks or priorities of objects. For example, in age estimation, $\theta_i \leq \theta_j$ means that people in age class $\theta_i$ look younger than those in $\theta_j$.

We may use the symbol $\to$, instead of $\leq$, to denote an order on a finite set $\Theta$. Then, the order can be represented by a directed graph (Gross & Yellen, 2006) using elements in $\Theta$ as nodes. If $\theta_i \to \theta_j$, there is a directed edge from node $\theta_i$ to node $\theta_j$. The order graph is acyclic because of antisymmetry and transitivity. For example, for $n, m \in \mathbb{N}$, let $n \to m$ denote that $m$ is a multiple of $n$. Note that it is an order on any subset of $\mathbb{N}$. Figure 1(a) is the graph representing this order on $\{1, \ldots, 9\}$.

Elements $\theta_i$ and $\theta_j$ are *comparable* if $\theta_i \to \theta_j$ or $\theta_j \to \theta_i$, or *incomparable* otherwise. In Figure 1(a), 6 and 8 are incomparable. In age estimation, it is difficult to compare apparent ages of people in different ethnic groups or of different genders.

An order on a set $\Theta$ is *total* (or *linear*) if all elements in $\Theta$ are comparable to one another. In such a case, $\Theta$ is called a linearly ordered set. In some real-world problems, orders are not linear. In this work, a subset $\Theta_c$ of $\Theta$ is referred to as a *chain*, if $\Theta_c$ is linearly ordered and also maximal, *i.e.* there is no proper superset of $\Theta_c$ that is linearly ordered. In Figure 1(a), nodes 1, 2, 4, and 8 form a chain. In Figure 1(b), the entire set is composed of three disjoint chains.

### 3.2    ORDER LEARNING – BASICS

Let $\Theta = \{\theta_1, \theta_2, \cdots, \theta_n\}$ be an ordered set of classes, where each class $\theta_i$ represents one or more object instances. For example, in age estimation, age class 11 is the set of 11-year-olds. The objective of order learning is to determine the order graph, such as Figure 1(a) or (b), and categorize an object instance into one of the classes. However, in many cases, order graphs are given explicitly or obvious from the contexts. For example, in quality assessment, there are typically five classes (poor $\to$ satisfactory $\to$ good $\to$ very good $\to$ excellent), forming a single chain. Also, in age estimation, suppose that an algorithm first classifies a person's gender into female or male and then estimates the age differently according to the gender. In this case, implicitly, there are separate age classes for each gender, and the age classes compose two disjoint chains similarly to Figure 1(b). Thus, in this subsection, we assume that the order graph is already known. Also, given an object instance, we assume that the chain to which the instance belongs is known. Then, we attempt to categorize the instance into one of the classes in the chain. Section 3.4 will propose the order learning in the case of an unknown order graph, composed of disjoint chains.

Instead of directly estimating the class of each instance, we learn pairwise ordering relationship between two instances. Let $\Theta_c = \{0, 1, \ldots, N-1\}$ be a chain, where $N$ is the number of classes. Let

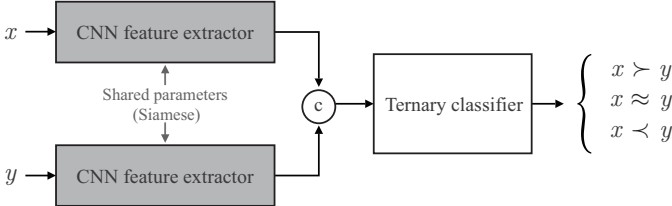

Figure 2: Illustration of the pairwise comparator, where ⓒ denotes concatenation.

$x$ and $y$ be two instances belonging to classes in $\Theta_c$. Let $\theta(\cdot)$ denote the class of an instance. Then, $x$ and $y$ are compared and their ordering relationship is defined according to their class difference as

$$x \succ y \qquad \text{if } \theta(x) - \theta(y) > \tau, \tag{1}$$
$$x \approx y \qquad \text{if } |\theta(x) - \theta(y)| \leq \tau, \tag{2}$$
$$x \prec y \qquad \text{if } \theta(x) - \theta(y) < -\tau, \tag{3}$$

where $\tau$ is a threshold. To avoid confusion, we use '$\succ, \approx, \prec$' for the instance ordering, while '$>$, $=$, $<$' for the class order. In practice, the categorization in (1)~(3) is performed by a pairwise comparator in Figure 2, which consists of a Siamese network and a ternary classifier (Lee & Kim, 2019b). To train the comparator, only comparable instance pairs are employed.

We estimate the class $\theta(x)$ of a test instance $x$ by comparing it with reference instances $y_m$, $0 \leq m \leq M - 1$, where $M$ is the number of references. The references are selected from training data such that they are from the same chain as $x$. Given $x$ and $y_m$, the comparator provides one of three categories '$\succ, \approx, \prec$' as a result. Let $\theta'$ be an estimate of the true class $\theta(x)$. Then, the consistency between the comparator result and the estimate is defined as

$$\phi_{\text{con}}(x, y_m, \theta') = \tag{4}$$
$$\big[x \succ y_m\big]\big[\theta' - \theta(y_m) > \tau\big] + \big[x \approx y_m\big]\big[|\theta' - \theta(y_m)| \leq \tau\big] + \big[x \prec y_m\big]\big[\theta' - \theta(y_m) < -\tau\big]$$

where $[\cdot]$ is the indicator function. The function $\phi_{\text{con}}(x, y_m, \theta')$ returns either 0 for an inconsistent case or 1 for a consistent case. For example, suppose that the pairwise comparator declares $x \prec y_m$ but $\theta' - \theta(y_m) > \tau$. Then, $\phi_{\text{con}}(x, y_m, \theta') = 0 \cdot 1 + 0 \cdot 0 + 1 \cdot 0 = 0$. Due to a possible classification error of the comparator, this inconsistency may occur even when the estimate $\theta'$ equals the true class $\theta(x)$. To maximize the consistency with all references, we estimate the class of $x$ by

$$\hat{\theta}_{\text{MC}}(x) = \arg\max_{\theta' \in \Theta_c} \sum_{m=0}^{M-1} \phi_{\text{con}}(x, y_m, \theta'), \tag{5}$$

which is called the maximum consistency (MC) rule. Figure 3 illustrates this MC rule.

It is noted that '$\succ, \approx, \prec$' is not an mathematical order. For example, if $\theta(x) + \frac{3}{4}\tau = \theta(y) = \theta(z) - \frac{3}{4}\tau$, then $x \approx y$ and $y \approx z$ but $x \prec z$. This is impossible in an order. More precisely, due to the quantization effect of the ternary classifier in (1)~(3), '$\succ, \approx, \prec$' is quasi-transitive (Sen, 1969), and '$\approx$' is symmetric but intransitive. We use this quasi-transitive relation to categorize an instance into one of the classes, on which a mathematical order is well defined.

## 3.3 ORDER LEARNING – SUPERVISED CHAINS

### 3.3.1 SINGLE-CHAIN HYPOTHESIS (1CH)

In the simplest case of 1CH, all classes form a single chain $\Theta_c = \{0, 1, \ldots, N-1\}$. For example, in 1CH age estimation, people's ages are estimated regardless of their ethnic groups or genders.

We implement the comparator in Figure 2 using CNNs, as described in Section 4.1. Let $\mathbf{q}^{xy} = (q_0^{xy}, q_1^{xy}, q_2^{xy})$ be the one-hot vector, indicating the ground-truth ordering relationship between training instances $x$ and $y$. Specifically, $(1, 0, 0)$, $(0, 1, 0)$, and $(0, 0, 1)$ represent $x \succ y$, $x \approx y$, and $x \prec y$. Also, $\mathbf{p}^{xy} = (p_0^{xy}, p_1^{xy}, p_2^{xy})$ is the corresponding softmax probability vector of the comparator. We train the comparator to minimize the comparator loss

$$\ell_{\text{co}} = -\sum_{x \in \mathcal{T}} \sum_{y \in \mathcal{R}} \sum_{j=0}^{2} q_j^{xy} \log p_j^{xy} \tag{6}$$

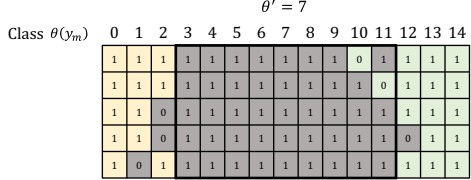

Figure 3: Consistency computation in the MC rule: It is illustrated how to compute the sum in (5) for two candidates $\theta' = 7$ and 9. Each box represents a reference $y_m$. There are 5 references for each class in $\{0, \ldots, 14\}$. Comparison results are color-coded (yellow for $x \succ y_m$, gray for $x \approx y_m$, and green for $x \prec y_m$). The bold black rectangle encloses the references satisfying $|\theta' - \theta(y_m)| \leq \tau$, where $\tau = 4$. The computed consistency $\phi_{\mathrm{con}}(x, y_m, \theta')$ in (5) is written within the box. For $\theta' = 7$, there are six inconsistent boxes. For $\theta' = 9$, there are 24 such boxes. In this example, $\theta' = 7$ minimizes the inconsistency, or equivalently maximizes the consistency. Therefore, $\hat{\theta}_{\mathrm{MC}}(x) = 7$.

where $\mathcal{T}$ is the set of all training instances and $\mathcal{R} \subset \mathcal{T}$ is the set of reference instances. First, we initialize $\mathcal{R} = \mathcal{T}$ and minimize $\ell_{\mathrm{co}}$ via the stochastic gradient descent. Then, we reduce the reference set $\mathcal{R}$ by sampling references from $\mathcal{T}$. Specifically, for each class in $\Theta_c$, we choose $M/N$ reference images to minimize the same loss $\ell_{\mathrm{co}}$, where $M$ is the number of all references and $N$ is the number of classes. In other words, the reliability score of a reference candidate $y$ is defined as

$$\alpha(y) = \sum_{x \in \mathcal{T}} \sum_{j=0}^{2} q_j^{xy} \log p_j^{xy} \tag{7}$$

and the $M/N$ candidates with the highest reliability scores are selected. Next, after fixing the reference set $\mathcal{R}$, the comparator is trained to minimize the loss $\ell_{\mathrm{co}}$. Then, after fixing the comparator parameters, the reference set $\mathcal{R}$ is updated to minimize the same loss $\ell_{\mathrm{co}}$, and so forth.

In the test phase, an input instance is compared with the $M$ references and its class is estimated using the MC rule in (5).

### 3.3.2 $K$-CHAIN HYPOTHESIS ($K$CH)

In $K$CH, we assume that classes form $K$ disjoint chains, as in Figure 1(b). For example, in the supervised 6CH for age estimation, we predict a person's age according to the gender in {female, male} and the ethnic group in {African, Asian, European}. Thus, there are 6 chains in total. In this case, people in different chains are assumed to be incomparable for age estimation. It is supervised, since gender and ethnic group annotations are used to separate the chains. The supervised 2CH or 3CH also can be implemented by dividing chains by genders only or ethnic groups only.

The comparator is trained similarly to 1CH. However, in computing the comparator loss in (6), a training instance $x$ and a reference $y$ are constrained to be from the same chain. Also, during the test, the type (or chain) of a test instance should be determined. Therefore, a $K$-way type classifier is trained, which shares the feature extractor with the comparator in Figure 2 and uses additional fully-connected (FC) layers. Thus, the overall loss is given by

$$\ell = \ell_{\mathrm{co}} + \ell_{\mathrm{ty}} \tag{8}$$

where $\ell_{\mathrm{co}}$ is the comparator loss and $\ell_{\mathrm{ty}}$ is the type classifier loss. The comparator and the type classifier are jointly trained to minimize this overall loss $\ell$.

During the test, given an input instance, we determine its chain using the type classifier, and compare it with the references from the same chain, and then estimate its class using the MC rule in (5).

### 3.4 ORDER LEARNING – UNSUPERVISED CHAINS

This subsection proposes an algorithm to separate classes into $K$ disjoint chains when there are no supervision or annotation data available for the separation. First, we randomly partition the training set $\mathcal{T}$ into $\mathcal{T}_0, \mathcal{T}_1, \ldots, \mathcal{T}_{K-1}$, where $\mathcal{T} = \mathcal{T}_0 \cup \ldots \cup \mathcal{T}_{K-1}$ and $\mathcal{T}_k \cap \mathcal{T}_l = \emptyset$ for $k \neq l$. Then, similarly

---

**Algorithm 1** Order Learning with Unsupervised Chains

---

**Input:** $\mathcal{T}$ = training set of ordinal data, $K$ = # of chains, $N$ = # of classes in each chain, and $M$ = # of references in each chain

1: Partition $\mathcal{T}$ randomly into $\mathcal{T}_0, \ldots, \mathcal{T}_{K-1}$ and train a pairwise comparator
2: **for** each chain $k$ **do**                                               ▷ *Reference Selection* $(\mathcal{R}_k)$
3:     From $\mathcal{T}_k$, select $M/N$ references $y$ with the highest reliability scores $\alpha_k(y)$
4: **end for**
5: **repeat**
6:     **for** each instance $x$ **do**                               ▷ *Membership Update* $(\mathcal{T}_k)$
7:         Assign it to $\mathcal{T}_{k^*}$, where $k^* = \arg\max_k \beta_k(x)$ subject to the regularization constraint
8:     **end for**
9:     Fine-tune the comparator and train a type classifier using $\mathcal{T}_0, \ldots, \mathcal{T}_{K-1}$ to minimize $\ell = \ell_{\text{co}} + \ell_{\text{ty}}$
10:     **for** each instance $x$ **do**                          ▷ *Membership Refinement* $(\mathcal{T}_k)$
11:         Assign it to $\mathcal{T}_{k'}$ where $k'$ is its type classification result
12:     **end for**
13:     **for** each chain $k$ **do**                                   ▷ *Reference Selection* $(\mathcal{R}_k)$
14:     From $\mathcal{T}_k$, select $M/N$ references $y$ with the highest reliability scores $\alpha_k(y)$
15:     **end for**
16: **until** convergence or predefined number of iterations

**Output:** Pairwise comparator, type classifier, reference sets $\mathcal{R}_0, \ldots, \mathcal{R}_{K-1}$

---

to (6), the comparator loss $\ell_{\text{co}}$ can be written as

$$\ell_{\text{co}} = -\sum_{k=0}^{K-1}\sum_{x\in\mathcal{T}_k}\sum_{y\in\mathcal{R}_k}\sum_{j=0}^{2} q_j^{xy}\log p_j^{xy} = -\sum_{k=0}^{K-1}\sum_{y\in\mathcal{R}_k}\alpha_k(y) = -\sum_{k=0}^{K-1}\sum_{x\in\mathcal{T}_k}\beta_k(x) \quad (9)$$

where $\mathcal{R}_k \subset \mathcal{T}_k$ is the set of references for the $k$th chain, $\alpha_k(y) = \sum_{x\in\mathcal{T}_k}\sum_j q_j^{xy}\log p_j^{xy}$ is the reliability of a reference $y$ in the $k$th chain, and $\beta_k(x) = \sum_{y\in\mathcal{R}_k}\sum_j q_j^{xy}\log p_j^{xy}$ is the affinity of an instance $x$ to the references in the $k$th chain. Note that $\beta_k(x) = -\sum_{y\in\mathcal{R}_k} D(\mathbf{q}^{xy}\|\mathbf{p}^{xy})$ where $D$ is the Kullback-Leibler distance (Cover & Thomas, 2006). Second, after fixing the chain membership $\mathcal{T}_k$ for each chain $k$, we select references $y$ to maximize the reliability scores $\alpha_k(y)$. These references form $\mathcal{R}_k$. Third, after fixing $\mathcal{R}_0, \ldots, \mathcal{R}_{K-1}$, we update the chain membership $\mathcal{T}_0, \ldots, \mathcal{T}_{K-1}$, by assigning each training instance $x$ to the $k$th chain that maximizes the affinity score $\beta_k(x)$. The second and third steps are iteratively repeated. Both steps decrease the same loss $\ell_{\text{co}}$ in (9).

The second and third steps are analogous to the centroid rule and the nearest neighbor rule in the $K$-means clustering (Gersho & Gray, 1991), respectively. The second step determines representatives in each chain (or cluster), while the third step assigns each instance to an optimal chain according to the affinity. Furthermore, both steps decrease the same loss alternately.

However, as described in **Algorithm 1**, we modify this iterative algorithm by including the membership refinement step in lines $10 \sim 12$. Specifically, we train a $K$-way type classifier using $\mathcal{T}_0, \ldots, \mathcal{T}_{K-1}$. Then, we accept the type classification results to refine $\mathcal{T}_0, \ldots, \mathcal{T}_{K-1}$. This refinement is necessary because the type classifier should be used in the test phase to determine the chain of an unseen instance. Therefore, it is desirable to select the references also after refining the chain membership. Also, in line 7, if we assign an instance $x$ to maximize $\beta_k(x)$ only, some classes may be assigned too few training instances, leading to data imbalance. To avoid this, we enforce the regularization constraint so that every class is assigned at least a predefined number of instances. This regularized membership update is described in Appendix A.

## 4 AGE ESTIMATION

We develop an age estimator based on the proposed order learning. Order learning is suitable for age estimation, since telling the older one between two people is easier than estimating each person's age directly (Chang et al., 2010; Zhang et al., 2017a).

### 4.1 IMPLEMENTATION DETAILS

It is less difficult to distinguish between a 5-year-old and a 10-year-old than between a 65-year-old and a 70-year-old. Therefore, in age estimation, we replace the categorization based on the

Table 1: A summary of the balanced dataset, formed from MORPH II, AFAD, and UTK. An element $\frac{n}{m}$ means that, out of $m$ images in the original dataset, $n$ images are sampled for the balanced dataset.

| | MORPH II | | AFAD | | UTK | | Balanced | |
|---|---|---|---|---|---|---|---|---|
| | Male | Female | Male | Female | Male | Female | Male | Female |
| African | $\frac{4,022}{36,772}$ | $\frac{4,446}{5,748}$ | $\frac{0}{0}$ | $\frac{0}{0}$ | $\frac{2,047}{2,319}$ | $\frac{1,871}{2,209}$ | 6,069 | 6,317 |
| Asian | $\frac{153}{153}$ | $\frac{17}{17}$ | $\frac{5,000}{100,752}$ | $\frac{5,000}{63,680}$ | $\frac{1,015}{1,575}$ | $\frac{1,200}{1,859}$ | 6,168 | 6,217 |
| European | $\frac{1,852}{7,992}$ | $\frac{2,602}{2,602}$ | $\frac{0}{0}$ | $\frac{0}{0}$ | $\frac{4,487}{5,477}$ | $\frac{3,437}{4,601}$ | 6,339 | 6,039 |

arithmetic difference in (1)∼(3) with that based on the geometric ratio as follows.

$$x \succ y \qquad \text{if } \log \theta(x) - \log \theta(y) > \tau_{\text{age}}, \tag{10}$$

$$x \approx y \qquad \text{if } |\log \theta(x) - \log \theta(y)| \leq \tau_{\text{age}}, \tag{11}$$

$$x \prec y \qquad \text{if } \log \theta(x) - \log \theta(y) < -\tau_{\text{age}}, \tag{12}$$

which represent 'older,' 'similar,' and 'younger.' The consistency in (4) is also modified accordingly.

There are 5 reference images for each age class within range $[15, 80]$ in this work ($M = 330, N = 66$). Thus, a test image should be compared with 330 references. However, we develop a two-step approach, which does at most 130 comparisons but performs as good as the method using 330 comparisons. The two-step estimation is employed in all experiments. It is described in Appendix B.

We align all facial images using SeetaFaceEngine (Zhang et al., 2014) and resize them into $256 \times 256 \times 3$. Then, we crop a resized image into $224 \times 224 \times 3$. For the feature extractors in Figure 2, we use VGG16 without the FC layers (Simonyan & Zisserman, 2014). They yield 512-channel feature vectors. Then, the vectors are concatenated and input to the ternary classifier, which has three FC layers, yielding 512-, 512-, and 3-channel vectors sequentially. The 3-channel vector is normalized to the softmax probabilities of the three categories '$\succ, \approx, \prec$.' In (10)∼(12), $\tau_{\text{age}}$ is set to 0.1.

In $K$CH with $K \geq 2$, the type (or chain) of a test image should be determined. Thus, we design a type classifier, which shares the feature extractor with the comparator. Similarly to the ternary classifier, the type classifier uses three FC layers, yielding 512-, 512-, and $K$-channel vectors sequentially. The comparator and the type classifier are jointly trained.

To initialize the feature extractors, we adopt the VGG16 parameters pre-trained on ImageNet (Deng et al., 2009). We randomly initialize all the other layers. We update the parameters using the Adam optimizer (Kingma & Ba, 2014). We set the learning rate to $10^{-4}$ for the first 70 epochs. Then, we select 5 references for each age class. Using the selected references, we fine-tune the network with a learning rate of $10^{-5}$. We repeat the reference selection and the parameter fine-tuning up to 3 times.

In the case of unsupervised chains, we enforce the regularization constraint (line 7 in **Algorithm 1**). By default, for each age, all chains are constrained to be assigned the same number of training images. If there are $L$ training images of $\theta$-year-olds, the age classes $\theta$ in the $K$ chains are assigned $L/K$ images, respectively, according to the affinity scores $\beta_k(x)$ by **Algorithm 2** in Appendix A.

## 4.2 DATASETS AND EVALUATION METRICS

MORPH II (Ricanek & Tesafaye, 2006) is the most popular age estimation benchmark, containing about 55,000 facial images in the age range $[16, 77]$. IMDB-WIKI (Rothe et al., 2018) is another dataset containing about 500,000 celebrity images obtained from IMDB and Wikipedia. It is sometimes used to pre-train age estimation networks. Optionally, we also select 150,000 clean data from IMDB-WIKI to pre-train the proposed pairwise comparator.

Although several facial age datasets are available, most are biased to specific ethnic groups or genders. Data unbalance restricts the usability and degrades the generalization performance. Thus, we form a 'balanced dataset' from MORPH II, AFAD (Niu et al., 2016), and UTK (Zhang et al., 2017b). Table 1 shows how the balanced dataset is organized. Before sampling images from MORPH II, AFAD, and UTK, we rectify inconsistent labels by following the strategy in Yip et al. (2018). For each combination of gender in {female, male} and ethnic group in {African, Asian, European}, we sample about 6,000 images. Also, during the sampling, we attempt to make the age distribution as

Table 2: Performance comparison on the MORPH II dataset: * means that the networks are pre-trained on IMDB-WIKI, and † the values are read from the reported CS curves or measured by experiments. The best results are boldfaced, and the second best ones are underlined.

| | Setting A | | Setting B | | Setting C (SE) | | Setting D (RS) | |
|---|---|---|---|---|---|---|---|---|
| | MAE | CS(%) | MAE | CS(%) | MAE | CS(%) | MAE | CS(%) |
| OHRank (Chang et al., 2011) | - | - | - | - | - | - | 6.07 | 56.3 |
| OR-CNN (Niu et al., 2016) | - | - | - | - | - | - | 3.27 | 73.0† |
| Ranking-CNN (Chen et al., 2017) | - | - | - | - | - | - | 2.96 | 85.0† |
| DMTL (Han et al., 2018) | - | - | - | - | 3.00 | 85.3 | - | - |
| DEX* (Rothe et al., 2018) | 2.68 | - | - | - | - | - | - | - |
| DRFs (Shen et al., 2018) | 2.91 | 82.9 | 2.98 | - | - | - | 2.17 | 91.3 |
| MO-CNN* (Tan et al., 2018) | 2.52 | 85.0† | 2.70 | 83.0† | - | - | - | - |
| MV (Pan et al., 2018) | - | - | - | - | 2.80 | 87.0† | 2.41 | 90.0† |
| MV* | - | - | - | - | 2.79 | - | **2.16** | - |
| BridgeNet* (Li et al., 2019) | **2.38** | 91.0† | **2.63** | 86.0† | - | - | - | - |
| Proposed (1CH) | 2.69 | 89.1 | 3.00 | 85.2 | 2.76 | 88.0 | 2.32 | 92.4 |
| Proposed* (1CH) | 2.41 | **91.7** | 2.75 | **88.2** | **2.68** | **88.8** | 2.22 | **93.3** |

uniform as possible within range $[15, 80]$. The balanced dataset is partitioned into training and test subsets with ratio $8 : 2$.

For performance assessment, we calculate the mean absolute error (MAE) (Lanitis et al., 2004) and the cumulative score (CS) (Geng et al., 2006). MAE is the average absolute error between predicted and ground-truth ages. Given a tolerance level $l$, CS computes the percentage of test images whose absolute errors are less than or equal to $l$. In this work, $l$ is fixed to 5, as done in Chang et al. (2011), Han et al. (2018), and Shen et al. (2018).

## 4.3 EXPERIMENTAL RESULTS

Table 2 compares the proposed algorithm (1CH) with conventional algorithms on MORPH II. As evaluation protocols for MORPH II, we use four different settings, including the 5-fold subject-exclusive (SE) and the 5-fold random split (RS) (Chang et al., 2010; Guo & Wang, 2012). Appendix C.1 describes these four settings in detail and provides an extended version of Table 2.

OHRank, OR-CNN, and Ranking-CNN are all based on ordinal regression. OHRank uses traditional features, yielding relatively poor performances, whereas OR-CNN and Ranking-CNN use CNN features. DEX, DRFs, MO-CNN, MV, and BridgeNet employ VGG16 as backbone networks. Among them, MV and BridgeNet achieve the state-of-the-art results, by employing the mean-variance loss and the gating networks, respectively. The proposed algorithm outperforms these algorithms in setting C, which is the most challenging task. Furthermore, in terms of CS, the proposed algorithm yields the best performances in all four settings. These outstanding performances indicate that order learning is an effective approach to age estimation.

In Table 3, we analyze the performances of the proposed algorithm on the balanced dataset according to the number of hypothesized chains. We also implement and train the state-of-the-art MV on the balanced dataset and provide its results using supervised chains.

Let us first analyze the performances of the proposed algorithm using 'supervised' chains. The MAE and CS scores on the balanced dataset are worse than those on MORPH II, since the balanced dataset contains more diverse data and thus is more challenging. By processing facial images separately according to the genders (2CH), the proposed algorithm reduces MAE by 0.05 and improves CS by 0.2% in comparison with 1CH. Similar improvements are obtained by 3CH or 6CH, which consider the ethnic groups only or both gender and ethnic groups, respectively. In contrast, in the case of MV, multi-chain hypotheses sometimes degrade the performances; *e.g.*, MV (6CH) yields a lower CS than MV (1CH). Regardless of the number of chains, the proposed algorithm trains a single comparator but uses a different set of references for each chain. The comparator is a ternary classifier. In contrast, MV (6CH) should train six different age estimators, each of which is a 66-way classifier, to handle different chains. Thus, their training is more challenging than that of the single ternary classifier. Note that, for the multi-chain hypotheses, the proposed algorithm first identifies the chain of a test image using the type classifiers, whose accuracies are about 98%. In Table 3, these

Table 3: Comparison of the proposed algorithm with MV on the balanced dataset. In MV and the supervised algorithm, multi-chain hypotheses divide data by the genders and/or the ethnic groups.

| | MAE | | | | CS(%) | | | |
|---|---|---|---|---|---|---|---|---|
| | 1CH | 2CH | 3CH | 6CH | 1CH | 2CH | 3CH | 6CH |
| MV (Pan et al., 2018) | 4.49 | 4.52 | 4.44 | 4.40 | 69.9 | 70.1 | 70.3 | 69.6 |
| Proposed (supervised) | 4.23 | 4.18 | 4.19 | 4.18 | 73.2 | 73.4 | 73.4 | 73.4 |
| Proposed (unsupervised) | - | **4.16** | 4.17 | **4.16** | - | **74.0** | 73.9 | **74.0** |

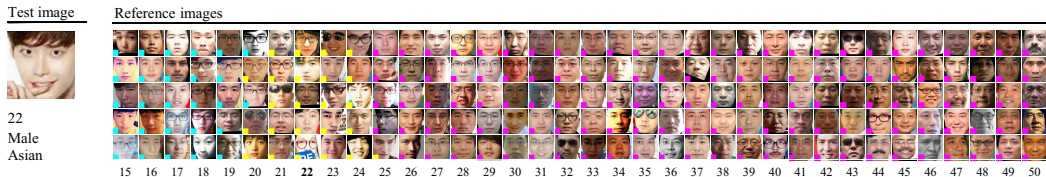

Figure 4: Age estimation in 6CH: Only the references of ages from 15 to 50 are shown. Comparison results are color-coded. Cyan, yellow, and magenta mean that the test subject is older than ($\succ$), similar to ($\approx$), and younger than ($\prec$) a reference. The age is estimated correctly as 22.

type classifiers are used to obtain the results of the proposed algorithm, whereas the ground-truth gender and ethnic group of each test image are used for MV.

Figure 4 shows how to estimate an age in 6CH. In this test, the subject is a 22-year-old Asian male. He is compared with the references who are also Asian males. Using the comparison results, the age is correctly estimated as 22 by the MC rule in (5).

Table 4 lists the MAE results for each test chain. Europeans yield poorer MAEs than Africans or Asians. However, this is not due to inherent differences between ethnic groups. It is rather caused by differences in image qualities. As listed in Table 1, more European faces are sampled from UTK. The UTK faces were crawled from the Internet and their qualities are relatively low. Also, from the cross-chain test results using 6CH, some observations can be made:

- Except for the As-F test chain, the lowest MAE is achieved by the references in the same chain.
- Eu-M and Eu-F are mutually compatible. For Eu-M, the second best performance is obtained by the Eu-F references, and vice versa. On the other hand, some chains, such as Af-M and Eu-F, are less compatible for the purpose of the proposed age estimation.

Table 3 also includes the performances of the proposed algorithm using 'unsupervised' chains. The unsupervised algorithm outperforms the supervised one, which indicates that the gender or ethnic group is not the best information to divide data for age estimation. As in the supervised case, 2CH, 3CH, and 6CH yield similar performances, which means that two chains are enough for the balanced set. Compared with MV (1CH), the unsupervised algorithm (2CH) improves the performances significantly, by 0.33 in terms of MAE and 4.1% in terms of CS.

Figure 5 shows how training images are divided into two chains in the unsupervised 2CH. During the membership update, for each age, each chain is regularized to include at least a certain percentage ($\kappa$) of the training images. In the default mode, the two chains are assigned the same number of images with $\kappa = 50\%$. However, Appendix C.3 shows that the performance is not very sensitive to $\kappa$. At $\kappa = 10\%$, MAE = 4.17 and CS = 73.7%. From Figure 5, we observe

- The division of the chains is not clearly related to genders or ethnic groups. Regardless of genders or ethnic groups, about half of the images are assigned to chain 1 and the others to chain 2.
- At $\kappa = 10\%$, chain 1 mostly consists of middle ages, while chain 2 of 10s, 20s, 60s, and 70s.
- At $\kappa = 50\%$, there is no such strong age-dependent tendency. But, for some combinations of gender, ethnic group, and age band, it is not equal division. For example, for Asian females, a majority of 40s are assigned to chain 1 but a majority of 50s and 60s are assigned to chain 2.

The unsupervised algorithm is designed to divide instances into multiple clusters when gender and ethnic group information is unavailable. As shown in Appendix C.3, different $\kappa$'s yield various clustering results. Surprisingly, these different clusters still outperform the supervised algorithm.

Table 4: Cross-chain tests on the balanced dataset (MAEs). For example, in 6CH, when African male references are used to estimate the ages of Asian females, the resultant MAE is 3.82.

| Method | Reference chain | Test chain | | | | | |
|---|---|---|---|---|---|---|---|
| | | Af-M | Af-F | As-M | As-F | Eu-M | Eu-F |
| 1CH | All | 3.87 | 3.82 | 3.98 | 3.79 | 5.21 | 4.69 |
| 6CH | African-Male | **3.85** | 3.79 | 4.03 | 3.82 | 5.50 | 5.00 |
| | African-Female | 4.02 | **3.65** | 4.18 | 3.85 | 5.42 | 5.02 |
| | Asian-Male | 3.97 | 3.75 | **3.97** | 3.81 | 5.48 | 4.87 |
| | Asian-Female | 4.06 | 3.78 | 4.05 | 3.78 | 5.69 | 4.89 |
| | European-Male | 3.99 | 3.71 | 4.02 | 3.80 | **5.13** | 4.66 |
| | European-Female | 4.45 | 3.79 | 4.11 | **3.77** | 5.21 | **4.65** |

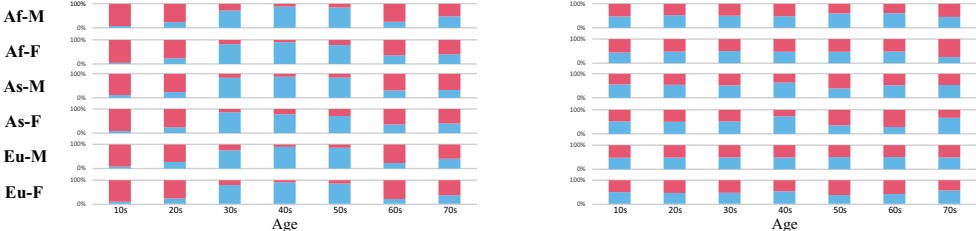

Figure 5: Distributions of training images in the unsupervised algorithm (2CH).

For example, at $\kappa = 10\%$, let us consider the age band of 20s and 30s. If the references in chain 2 are used to estimate the ages of people in chain 1, the average error is $4.6$ years. On the contrary, if the references in chain 1 are used for chain 2, the average error is $-5.4$ years. These opposite biases mean that people in chain 1 tend to look older than those in chain 2. These 'looking-older' people in 20s and 30s compose the blue cluster (chain 1) together with most people in 40s and 50s in Figure 5. In this case, 'looking-older' people in 20s and 30s are separated from 'looking-younger' ones by the unsupervised algorithm. This is more effective than the gender-based or ethnic-group-based division of the supervised algorithm. Appendix C presents more results on age estimation.

## 5 CONCLUSIONS

Order learning was proposed in this work. In order learning, classes form an ordered set, and each class represents object instances of the same rank. Its goal is to determine the order graph of classes and classify a test instance into one of the classes. To this end, we designed the pairwise comparator to learn ordering relationships between instances. We then decided the class of an instance by comparing it with reference instances in the same chain and maximizing the consistency among the comparison results. For age estimation, it was shown that the proposed algorithm yields the state-of-the-art performance even in the case of the single-chain hypothesis. The performance is further improved when the order graph is divided into multiple disjoint chains.

In this paper, we assumed that the order graph is composed of disjoint chains. However, there are more complicated graphs, *e.g.* Figure 1(a), than disjoint chains. For example, it is hard to recognize an infant's sex from its facial image (Porter et al., 1984). But, after puberty, male and female take divergent paths. This can be reflected by an order graph, which consists of two chains sharing common nodes up to a certain age. It is an open problem to generalize order learning to find an optimal order graph, which is not restricted to disjoint chains.

## ACKNOWLEDGEMENTS

This work was supported by 'The Cross-Ministry Giga KOREA Project' grant funded by the Korea government (MSIT) (No. GK19P0200, Development of 4D reconstruction and dynamic deformable action model based hyperrealistic service technology), and by the National Research Foundation of Korea (NRF) grant funded by the Korea government (MSIP) (No. NRF-2018R1A2B3003896).

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

## A    REGULARIZED MEMBERSHIP UPDATE

During the chain membership update in **Algorithm 1**, we assign an instance $x$ to chain $k$ to maximize $\beta_k(x)$ subject to the regularization constraint. As mentioned in Section 4.1, in age estimation, this regularization is enforced for each age. Let $\mathcal{X}$ denote the set of $\theta$-year-olds for a certain $\theta$. Also, let $\mathcal{K} = \{0, 1, \ldots, K-1\}$ be the set of chains. Suppose that we should assign at least a certain number ($L$) of instances in $\mathcal{X}$ to each chain. This is done by calling RegularAssign($\mathcal{K}$, $\mathcal{X}$, $L$) in **Algorithm 2**, which is a recursive function. **Algorithm 2** yields the membership function $c(x)$ as output. For example, $c(x) = 1$ means that $x$ belongs to chain 1.

---

**Algorithm 2** RegularAssign($\mathcal{K}$, $\mathcal{X}$, $L$)

---

**Input:** $\mathcal{K}$ = set of chains, $\mathcal{X}$ = set of instances, and $L$ = minimum number

1: **for** each $k \in \mathcal{K}$ **do**                                          ▷ Initialize chains
2:        $\mathcal{X}_k = \emptyset$
3: **end for**
4: **for** each $x \in \mathcal{X}$ **do**                                          ▷ Irregular partitioning
5:        $c(x) = \arg\max_{k \in \mathcal{K}} \beta_k(x)$
6:        $\mathcal{X}_{c(x)} = \mathcal{X}_{c(x)} \cup \{x\}$
7: **end for**
8: $k_m = \arg\min_{k \in \mathcal{K}} |\mathcal{X}_k|$                             ▷ Chain of the minimum size
9: **if** $|\mathcal{X}_{k_m}| \geq L$ **then**
10:      **return**
11: **else**
12:      $\mathcal{X} = \mathcal{X} - \mathcal{X}_{k_m}$
13:      **while** $|\mathcal{X}_{k_m}| < L$ **do**                               ▷ Increase $\mathcal{X}_{k_m}$
14:          $x' = \max_{x \in \mathcal{X}} \beta_{k_m}(x)$
15:          $\mathcal{X} = \mathcal{X} - \{x'\}$
16:          $\mathcal{X}_{k_m} = \mathcal{X}_{k_m} \cup \{x'\}$
17:      **end while**
18:      RegularAssign($\mathcal{K} - \{k_m\}$, $\mathcal{X}$, $L$)                ▷ Recursion
19: **end if**

**Output:**  Membership function $c(x)$

---

## B    TWO-STEP ESTIMATION

There are 5 reference images for each age within range $[15, 80]$ in this work. Thus, for the age estimation of a test image using the MC rule in (5), the test image should be compared with $M = 330$ reference images. However, we reduce the number of comparisons using a two-step approach. First, the test image is compared with the 35 references of ages $15, 25, \ldots, 75$ only, and a rough age estimate $\hat{\theta}_1$ is obtained using the MC rule. Second, it is compared with the 105 references of all ages within $[\hat{\theta}_1 - 10, \hat{\theta}_1 + 10]$, and the final estimate $\hat{\theta}_2$ is obtained. Since there are at least 10 common references in the first and second steps, the two-step estimation requires at most 130 comparisons.

## C  MORE EXPERIMENTS

### C.1  PERFORMANCE COMPARISON ON MORPH II

Four experimental settings are used for performance comparison on MORPH II (Ricanek & Tesafaye, 2006).

- Setting A: 5,492 images of Europeans are randomly selected and then divided into training and testing sets with ratio 8:2 (Chang et al., 2011).
- Setting B: About 21,000 images are randomly selected, while restricting the ratio between Africans and Europeans to 1:1 and that between females and males to 1:3. They are divided into three subsets (S1, S2, S3). The training and testing are done under two sub-settings (Guo & Mu, 2011).
  - (B1) training on S1, testing on S2 + S3
  - (B2) training on S2, testing on S1 + S3
- Setting C (SE): The entire dataset is randomly split into five folds, subject to the constraint that the same person's images should belong to only one fold, and the 5-fold cross-validation is performed.
- Setting D (RS): The entire dataset is randomly split into five folds without any constraint, and the 5-fold cross-validation is performed.

Table 5 is an extended version of Table 2. It includes the results of more conventional algorithms.

Table 5: Performance comparison on the MORPH II dataset: * means that the networks are pre-trained on IMDB-WIKI, and † the values are read from the reported CS curves or measured by experiments. The best results are boldfaced, and the second best ones are underlined.

| | Setting A | | Setting B | | Setting C (SE) | | Setting D (RS) | |
|---|---|---|---|---|---|---|---|---|
| | MAE | CS(%) | MAE | CS(%) | MAE | CS(%) | MAE | CS(%) |
| RED-SVM (Chang et al., 2010) | - | - | - | - | - | - | 6.49 | 49.0† |
| OHRank (Chang et al., 2011) | - | - | - | - | - | - | 6.07 | 56.3 |
| KPLS (Guo & Mu, 2011) | - | - | 4.18 | - | - | - | - | - |
| CPLF (Yi et al., 2014) | - | - | 3.63 | - | - | - | - | - |
| Huerta et al. (Huerta et al., 2015) | - | - | - | - | 3.88 | - | - | - |
| OR-CNN (Niu et al., 2016) | - | - | - | - | - | - | 3.27 | 73.0† |
| Tan et al. (Tan et al., 2016) | - | - | 3.03 | - | - | - | - | - |
| Ranking-CNN (Chen et al., 2017) | - | - | - | - | - | - | 2.96 | 85.0† |
| DMTL (Han et al., 2018) | - | - | - | - | 3.00 | 85.3 | - | - |
| DEX (Rothe et al., 2018) | 3.25 | - | - | - | - | - | - | - |
| DEX* | 2.68 | - | - | - | - | - | - | - |
| CMT (Yoo et al., 2018) | - | - | - | - | 2.91 | - | - | - |
| DRFs (Shen et al., 2018) | 2.91 | 82.9 | 2.98 | - | - | - | 2.17 | 91.3 |
| MO-CNN (Tan et al., 2018) | 2.93 | 83.0† | 2.86 | 82.0† | - | - | - | - |
| MO-CNN* | 2.52 | 85.0† | 2.70 | 83.0† | - | - | - | - |
| MV (Pan et al., 2018) | - | - | - | - | 2.80 | 87.0† | 2.41 | 90.0† |
| MV* | - | - | - | - | 2.79 | - | **2.16** | - |
| C3AE (Zhang et al., 2019) | - | - | - | - | - | - | 2.78 | - |
| C3AE* | - | - | - | - | - | - | 2.75 | - |
| BridgeNet* (Li et al., 2019) | **2.38** | 91.0† | **2.63** | 86.0† | - | - | - | - |
| Proposed (1CH) | 2.69 | 89.1 | 3.00 | 85.2 | 2.76 | 88.0 | 2.32 | 92.4 |
| Proposed* (1CH) | 2.41 | **91.7** | 2.75 | **88.2** | **2.68** | **88.8** | 2.22 | **93.3** |

## C.2 GENERALIZATION PERFORMANCE OF COMPARATOR ON FG-NET

We assess the proposed age estimator (1CH) on the FG-NET database (Panis et al., 2016). FG-NET is a relatively small dataset, composed of 1,002 facial images of 82 subjects. Ages range from 0 to 69. For FG-NET, the leave one person out (LOPO) approach is often used for evaluation. In other words, to perform tests on each subject, an estimator is trained using the remaining 81 subjects. Then, the results are averaged over all 82 subjects.

In order to assess the generalization performance, we do not retrain the comparator on the FG-NET data. Instead, we fix the comparator trained on the balanced dataset and just select references from the remaining subjects' faces in each LOPO test. For the comparator, the arithmetic scheme in (1)$\sim$(3) is tested as well as the default geometric scheme in (10)$\sim$(12).

For comparison, MV (Pan et al., 2018) is tested, but it is trained for each LOPO test.

Table 6 summarizes the comparison results. MV provides better average performances on the entire age range $[0, 69]$ than the proposed algorithm does. This is because the balanced dataset does not include subjects of ages between 0 and 14. If we reduce the test age range to $[15, 69]$, the proposed algorithm outperforms MV, even though the comparator is not retrained. These results indicate that the comparator generalizes well to unseen data, as long as the training images cover a desired age range. Also, note that the geometric scheme provides better performances than the arithmetic scheme.

Table 6: Performance comparison on FG-NET. The average performances over test ages within ranges $[0, 69]$ and $[15, 69]$ are reported, respectively.

|  | 0 to 69 | | 15 to 69 | |
|---|---|---|---|---|
|  | MAE | CS(%) | MAE | CS(%) |
| MV | 3.98 | 79.5 | 6.00 | 63.7 |
| Proposed (1CH, Geometric $\tau_{\mathrm{age}} = 0.15$) | 8.04 | 41.4 | 4.90 | 64.3 |
| Proposed (1CH, Arithmetic $\tau = 7$) | 9.26 | 33.1 | 5.32 | 64.1 |

Figure 6 compares MAEs according to a test age. Again, within the covered range $[15, 69]$, the proposed algorithm significantly outperforms MV especially when test subjects are older than 45.

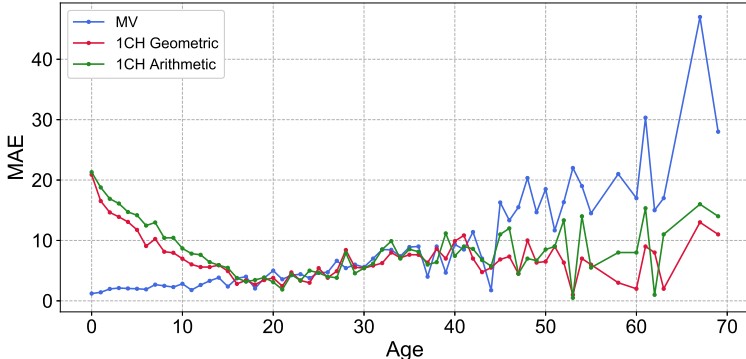

Figure 6: MAEs of the proposed algorithm (1CH) and MV on FG-NET in terms of a test age.

## C.3 PERFORMANCE ACCORDING TO $\kappa$

Table 7: MAE and CS performances of the unsupervised algorithm (2CH) on the balanced dataset, according to the minimum percentage ($\kappa$) constraint during the regularized membership update. The performances are not very sensitive to $\kappa$. The best performances are achieved in the default mode, *i.e.* at $\kappa = 50\%$.

| $\kappa$ (%) | 10 | 20 | 30 | 40 | 50 |
|---|---|---|---|---|---|
| MAE | 4.17 | 4.18 | 4.17 | 4.16 | 4.16 |
| CS (%) | 73.7 | 73.6 | 73.6 | 73.7 | 74.0 |

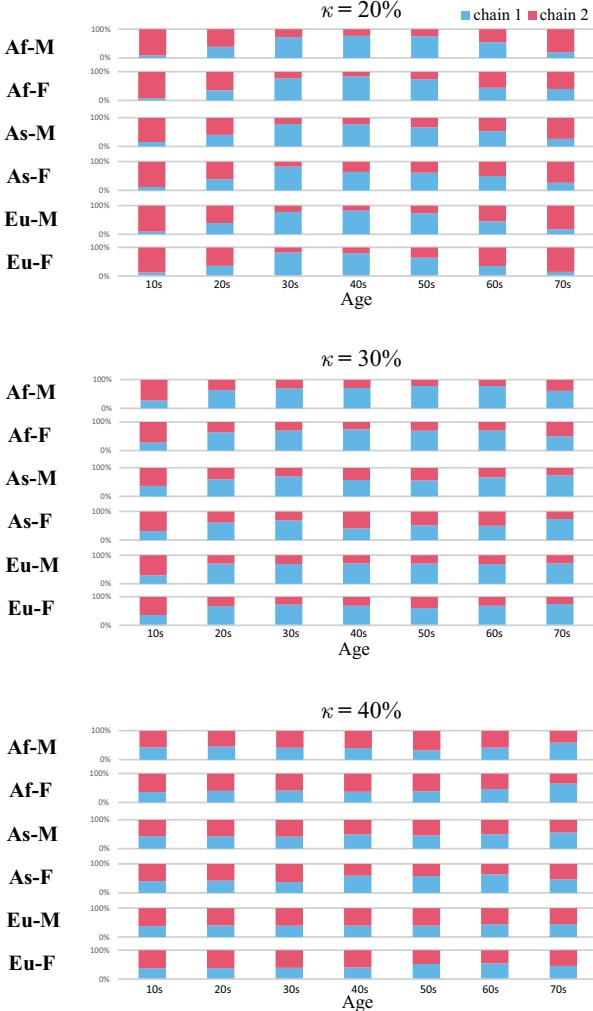

Figure 7: Distributions of training images in the unsupervised algorithm (2CH) at $\kappa = 20\%$, 30%, and 40%. From Figures 5 and 7, we see that stronger age-dependent tendencies are observed, as $\kappa$ gets smaller.

## C.4 Performance according to thresholds $\tau$ and $\tau_{\mathrm{age}}$

The ordering relationship between two instances can be categorized via the arithmetic scheme in (1)∼(3) using a threshold $\tau$ or the geometric scheme in (10)∼(12) using a threshold $\tau_{\mathrm{age}}$. Table 8 lists the performances of the proposed algorithm (1CH) according to these thresholds. We see that the geometric scheme outperforms the arithmetic scheme in general. The best performance is achieved with $\tau_{\mathrm{age}} = 0.1$, which is used in all experiments in the main paper. Note that the scores are poorer than those in Table 3, since the comparator is trained for a smaller number of epochs to facilitate this test. At $\tau_{\mathrm{age}} = 0.1$, two teenagers are declared to be not 'similar to' each other if their age difference is larger than about 1. Also, two forties are not 'similar' if the age difference is larger than about 5.

Table 8: The performances of the proposed algorithm (1CH) on the balanced dataset according to thresholds $\tau$ and $\tau_{\mathrm{age}}$.

|  | $\tau$ for arithmetic scheme | | | | | $\tau_{\mathrm{age}}$ for geometric scheme | | | | |
|  | 0 | 2 | 5 | 7 | 9 | 0.05 | 0.10 | 0.15 | 0.20 | 0.25 |
| --- | --- | --- | --- | --- | --- | --- | --- | --- | --- | --- |
| MAE | 4.42 | 4.36 | 4.33 | 4.32 | 4.33 | 4.38 | **4.31** | 4.32 | 4.41 | 4.41 |
| CS(%) | 71.0 | 71.7 | 72.2 | 72.2 | 72.5 | 71.4 | **72.8** | 72.2 | 71.8 | 71.7 |

## C.5 Performance according to number of references

Table 9: The performances of the proposed algorithm (supervised) on the balanced dataset according to the number of references for each age class ($M/N$). In general, the performances get better with more references. However, the performances are not very sensitive to $M/N$. They saturate when $M/N \geq 5$. Therefore, we set $M/N = 5$ in this work.

|  | 1CH | | 2CH | | 3CH | | 6CH | | Average | |
| $M/N$ | MAE | CS(%) | MAE | CS(%) | MAE | CS(%) | MAE | CS(%) | MAE | CS(%) |
| --- | --- | --- | --- | --- | --- | --- | --- | --- | --- | --- |
| 1 | 4.321 | 72.43 | 4.180 | 72.98 | 4.199 | 73.20 | 4.168 | 73.76 | 4.217 | 73.09 |
| 2 | 4.318 | 72.43 | 4.182 | 73.00 | 4.200 | 73.23 | 4.170 | 73.64 | 4.218 | 73.08 |
| 3 | 4.313 | 72.61 | 4.175 | 73.04 | 4.200 | 73.29 | 4.170 | 73.68 | 4.214 | 73.16 |
| 4 | 4.311 | 72.58 | 4.178 | 72.96 | 4.197 | 73.24 | 4.176 | 73.62 | 4.215 | 73.10 |
| 5 | 4.309 | 72.61 | 4.177 | 73.02 | 4.197 | 73.27 | 4.168 | 73.72 | 4.213 | 73.16 |
| 6 | 4.308 | 72.66 | 4.178 | 73.01 | 4.195 | 73.20 | 4.170 | 73.76 | 4.213 | 73.16 |
| 7 | 4.308 | 72.70 | 4.179 | 73.00 | 4.196 | 73.24 | 4.167 | 73.81 | 4.213 | 73.19 |
| 8 | 4.306 | 72.69 | 4.178 | 72.94 | 4.196 | 73.31 | 4.172 | 73.71 | 4.213 | 73.16 |
| 9 | 4.305 | 72.63 | 4.180 | 73.04 | 4.194 | 73.36 | 4.172 | 73.72 | 4.213 | 73.19 |
| 10 | 4.305 | 72.65 | 4.180 | 73.07 | 4.193 | 73.35 | 4.173 | 73.75 | 4.213 | 73.21 |

## C.6 Reference images

Figure 8 shows all references in the supervised 6CH.

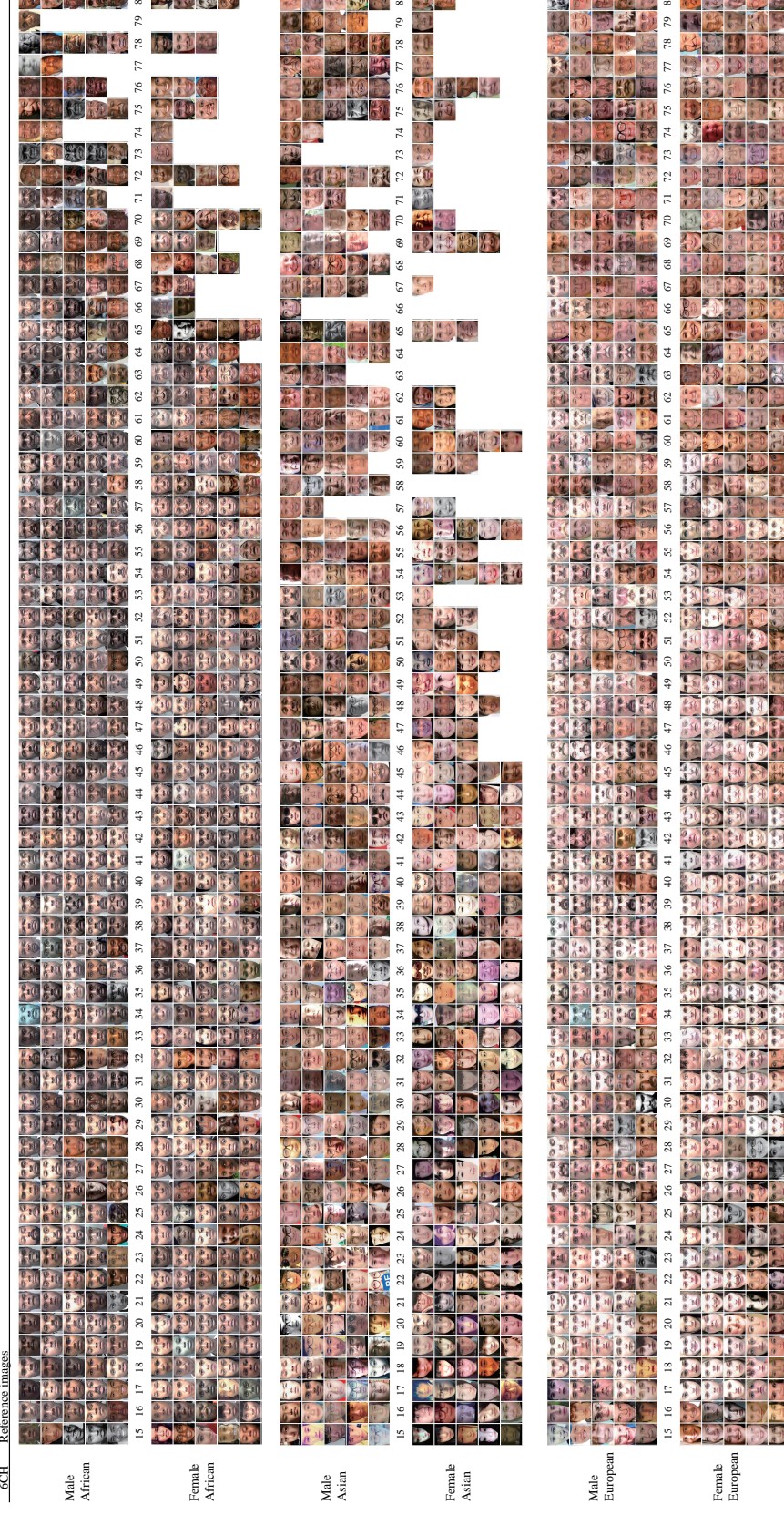

Figure 8: All reference images in the supervised 6CH. For some ages in certain chains, the balanced dataset includes less than 5 faces. In such cases, there are less than 5 references.

## C.7   AGE ESTIMATION EXAMPLES

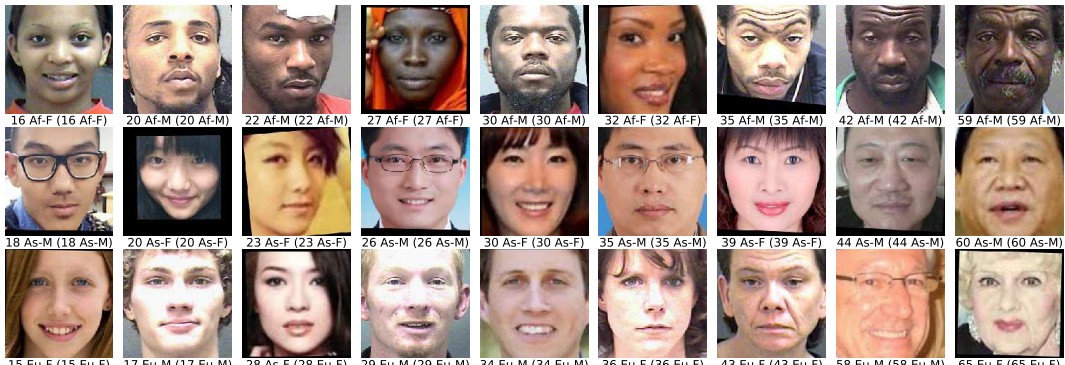

(a) Success cases

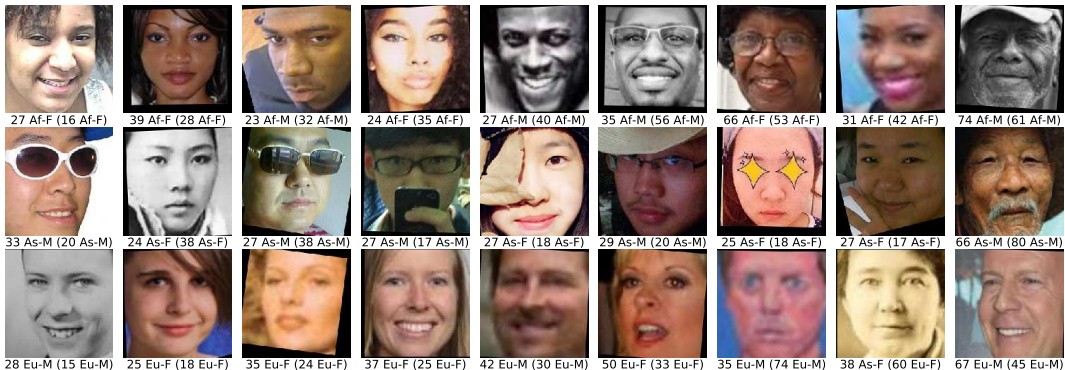

(b) Failure cases

Figure 9: Age estimation results of the proposed algorithm (supervised 6CH). For each face, the estimated label is provided together with the ground-truth in parentheses. In (a), the ages are estimated correctly. In the last row, third column, the ethnic group is misclassified. This happens rarely. In (b), failure cases are provided. These are hard examples due to various challenging factors, such as low quality photographs and occlusion by hairs, hats, hands, and stickers.

