# OpenReview forum: "Order Learning and Its Application to Age Estimation"
_ICLR.cc/2020/Conference — Accept (Poster)_

### Official Review · AnonReviewer1 · 2019-10-23
**Official Blind Review #1**

**Rating:** 8

**Review:**

The paper proposes a method for learning partial orders based on learning fuzzy pairwise comparisons (smaller/greater/approximatively equal to), and the retaining of a set of representants in each chain in order to allow consistent result by consistency maximization. The method is applied to the problem of estimating the age based on faces. Extensions proposed are the learning of multiple disjoint chains on this dataset, manually partitioned or learned through an iterative assignment algorithm that has similarities with a soft expectation-maximization principle. Extensive experiments are done on the age estimation problem with comparison to exisiting approaches, and an application to aesthetic assessment is proposed in appendix.

The paper is very well written and contains very extensive and relevant experiments. I only have minor concerns:

1) I couldn't find the explanation for the underlined numbers in Table 2.
2) It would be good to include the details of Appendix C in the text, as is, it looks like an artificial space saving trick.
3) typo in Section 3.2: "we assume that the chain, to which the instance belong, is known" -> "we assume that the chain to which the instance belongs is known"

**Experience Assessment:**

I do not know much about this area.

**Review Assessment: Checking Correctness Of Derivations And Theory:**

I carefully checked the derivations and theory.

**Review Assessment: Checking Correctness Of Experiments:**

I carefully checked the experiments.

**Review Assessment: Thoroughness In Paper Reading:**

I read the paper thoroughly.

---

> ### Author Response · Authors · 2019-11-08
> **Responses to Review #1**
>
> Thank you for your favorable review. We do appreciate it. Please find our responses to your minor concerns below.
>
> 1) Explanation for the underlined numbers in Table 2.
> => In each test, the second best result is underlined, while the best is boldfaced. We have explicitly stated this in the caption of Table 2 in the revised manuscript.
>
> 2) It would be good to include the details of Appendix C in the text
> => We agree with the reviewer. However, the inclusion needs significant changes in other parts of the paper because of the tight 10-page limit. If accepted, we will figure it out to include the details in the text.
>
> 3) Typo in Section 3.2
> => This has been corrected in the revised manuscript.

---

### Official Review · AnonReviewer3 · 2019-10-24
**Official Blind Review #3**

**Rating:** 6

**Review:**

This paper presents an order learning method and applies it to age estimation from facial images. It designs a pairwise comparator that categorizes ordering relationship between two instances into ternary classes of greater than, similar, and smaller than. Instead of directly estimating the class of each instance, it learns pairwise ordering relationship between two instances.

For age estimation from facial images, it uses a Siamese network as a feature extractor of the pairwise comparator, concatenates feature maps of dual inputs (test instance and one of the multiple reference instances that are selected from the training data) and applies it to a ternary classifier. Given the softmax probability scores of the classifier results, it estimates the final class as the one that maximizes a consistency rule computed over indicator functions. The comparator is trained to minimize a comparator loss computed over the softmax probability scores.

The paper also provides an extended version for multiple disjoint chains, where each chain may correspond to a higher-level attribute class, for example, gender or ethnic group. When there is no supervision available, it randomly partitions the training set.  and iteratively updates reliability scores and chain memberships.

Novelty-wise, I consider that the proposed solution to be satisfactorily innovative and on a different vein than the existing methods.

One concern about the method is that it imposes the geometric ratio (log distance) between the class distances in age estimation, considering the difference between 5 and 10-year-old instances is easier to detect than 65 to 70-year-old instances as stated in the paper. However, as far as it is understood from the provided discussion compared SOTA methods are not retrained or fine-tuned in the same manner. This raises the question of whether the slight performance improvement is a result of the geometric ratio, in particular when computing the cumulative score CS.

Another concern is that the performance is not stellar as the presented method underperforms in comparison to DRF (Shen et al. 2018) on MORPH II. The presentation of the results for FG-Net is neither complete nor included in the main paper, and results for CLAP2016 are missing. Discussion of the results Table 8 does not seem fair as it is not clear whether MV is retrained keeping the geometric ratio in mind. It might be the case that even for 15-69, MV might attain lover MAEs.

The paper also misses the latest SOTA, for instance, BridgeNet [1] is mentioned in a sentence, yet its results are omitted from the Tables. BridgeNet, according to its results, is on par with the proposed method. Also, the numbers reported in this paper and the numbers in [1] have a discrepancy, which causes confusion.

[1] Wanhua Li et al. Bridgenet: A continuity aware probabilistic network for age estimation, CVPR 2019.

**Experience Assessment:**

I have read many papers in this area.

**Review Assessment: Checking Correctness Of Derivations And Theory:**

I carefully checked the derivations and theory.

**Review Assessment: Checking Correctness Of Experiments:**

I carefully checked the experiments.

**Review Assessment: Thoroughness In Paper Reading:**

I read the paper thoroughly.

---

> ### Author Response · Authors · 2019-11-08
> **Responses to Review #3**
>
> Thank you for your constructive review. We do appreciate it. Please find our responses below.
>
> <Comment 1> One concern about the method is that it imposes the geometric ratio (log distance) between the class distances in age estimation... However, as far as it is understood from the provided discussion compared SOTA methods are not retrained or fine-tuned in the same manner. This raises the question of whether the slight performance improvement is a result of the geometric ratio...
> ==> Whereas the proposed algorithm compares two ages, conventional algorithms do not make such comparisons explicitly. Thus, it is not straightforward to modify conventional algorithms to use the geometric ratio. Maybe, their loss functions can be modified to penalize errors around young ages more severely. But this is outside the scope of our work.
> ==> The geometric scheme provides better performance than the arithmetic scheme in Eq. (1)~(3). Please note that we already compared the geometric scheme with the arithmetic scheme in Table 8 in Appendix D.3. As mentioned there, the scores are poorer than those in Table 3 since the comparator is trained for a smaller number of epochs. If we train the arithmetic scheme comparator with $\tau=7$ sufficiently, it yields MAE = 4.25 and CS=73.0. Thus, we have
>     ____________________________________________________
>     MV                    	                MAE=4.49      CS=69.9
>     Geometric scheme   	MAE=4.23      CS=73.2
>     Arithmetic scheme  	MAE=4.25      CS=73.0
>     ____________________________________________________
> The arithmetic scheme also outperforms MV, which indicates that the improvement is not just because of the geometric ratio. We will clarify this in the camera-ready.
>
>
> <Comment 2> Another concern is that the performance is not stellar as the presented method underperforms in comparison to DRF (Shen et al. 2018) on MORPH II. The presentation of the results for FG-Net is neither complete nor included in the main paper, and results for CLAP2016 are missing. Discussion of the results Table 8 does not seem fair as it is not clear whether MV is retrained keeping the geometric ratio in mind. It might be the case that even for 15-69, MV might attain lover MAEs.
> ==> We agree that the performance is not stellar on MORPH II. However, our focus is to propose order learning and apply the new ideas of reference selection and multiple chains to age estimation. To this end, we form the balanced dataset and carry out many experiments on it, instead of optimizing the proposed algorithm on an existing dataset. Table 3 shows that these ideas significantly improve the performances on the balanced dataset, as compared with MV, which is one of the SOTA algorithms.
> ==> To perform experiments on the CLAP2016 dataset, conventional algorithms typically use IMDB-WIKI for pre-training the networks. We will perform experiments on CLAP2016, including this pre-training. However, IMDB-WIKI is a large dataset with more than 500K images. We might not finish this experiment before the rebuttal due. If done before the due, we will let you know the results.
> ==> For the FG-Net experiment, we have revised the manuscript to include the performances of the arithmetic scheme, as well as the default geometric scheme. Again, the arithmetic scheme also outperforms MV for the age range [15, 69]. Please see the revised Appendix D.1 on page 16.
>
>
> < Comment 3> The paper also misses the latest SOTA, for instance, BridgeNet [1] is mentioned in a sentence, yet its results are omitted from the Tables. BridgeNet, according to its results, is on par with the proposed method. Also, the numbers reported in this paper and the numbers in [1] have a discrepancy, which causes confusion.
> ==> For the discrepancy, please note that there are many experimental settings on MORPH II, including
>
>     (A) Select 5,492 images of people of Caucasian descent. Randomly divide them into training (80%) and testing (20%) sets.
>     (B) Randomly select about 21,000 images, while restricting the ratio between Black and White to 1:1 and that between Female and Male to 1:3. Then, divide them into three subsets (S1, S2, S3). The training and testing are done under two sub-settings: (B1) training on S1, testing on S2 + S3, (B2) training on S2, testing on S1 + S2.
>     (C) Split the total dataset randomly into five folds and perform the 5-fold cross-validation. This setting is called  RS in the paper.
>     (D) Split the total dataset randomly into five folds, but the same person’s images should belong to only one fold. This is called SE in the paper.
>
> Settings A and B are used in BridgeNet [1], whereas C and D are used in this paper. Therefore, the reported numbers cannot be directly compared.
> ==> We will attempt to train the proposed algorithm on settings A and B and let you know if the results are obtained before the due date.

---

> > ### Author Response · Authors · 2019-11-15
> > **Comparison with BridgeNet**
> >
> > We are training the proposed algorithm (1CH) on MORPH II using setting A. The training is not complete yet, but its current results are as follows.
> >
> >    ----------------------------------------------------------
> >    BridgeNet              MAE=2.38	    CS=91.0%*
> >    Proposed(1CH)     MAE=2.44	    CS=91.2%
> >    ----------------------------------------------------------
> >    * the value is read from the reported CS curve
> >
> > The proposed algorithm provides comparable performance to BridgeNet. It yields a slightly worse MAE but a slightly better CS score. After completing the training, we will report the results in the revised paper.
> >
> >
> >
> > Thank you again for all your constructive comments.

---

### Official Review · AnonReviewer4 · 2019-11-12
**Official Blind Review #4**

**Rating:** 6

**Review:**

This paper departs from traditional age estimation methods by proposing an ordering/ranking way. A testing face is compared to those anchor faces from each age category to estimate the right category that the testing face belongs to. It is novel to use the MC rule to drive the order learning. What more promising is the unsupervised chains, which could automatically search for a more optimal multi-chain division scheme than the pre-defined data division. The paper is well written.

I have a few concerns as the following:
1.	The geometric ratio (Eq. 10) further enlarges the distance between younger faces and shortens the distance between older faces. This may improve the prediction of younger ages. However, this makes the prediction on older ages more confusion, i.e., too robust to distinguish adjacent categories. The results shown in Table 6 show a better number for Geometric, but it is tested on FG-NET which presents unbalanced age distribution (ages up to 40 years are the most populated). Therefore, the improvement may be caused by the data distribution that is more suitable for geometric ratio, i.e., more younger faces and less older faces. Table 8 also provides such comparison, but please clarify the testing dataset.
2.	In Algorithm 1, are you using a pre-trained comparator? If not, the T_k obtained from line 3 is still close to random. Such random initialization works for multiple chains with small mutual shift (the distance of references between chains). As indicated in Table 3, more chains will not significantly affect performance, which implies a relatively small mutual shift. However, if encounter a large shift, e.g., other properties would significantly affect the target property that you want to classify (ranking soccer teams where gender makes difference), will the proposed unsupervised learning converge?
3.	It would be great if the authors could provide more insight to the 2CH unsupervised results. How the two chains are divided by analyzing the trained model? What is the key to divide the data? Maybe, it is like a black-box that is hard to explain but this will not affect the merit of this paper.


**Experience Assessment:**

I have published one or two papers in this area.

**Review Assessment: Checking Correctness Of Derivations And Theory:**

I carefully checked the derivations and theory.

**Review Assessment: Checking Correctness Of Experiments:**

I carefully checked the experiments.

**Review Assessment: Thoroughness In Paper Reading:**

I read the paper thoroughly.

---

> ### Author Response · Authors · 2019-11-15
> **Responses to Review #4**
>
> Thank you for your positive review. We do appreciate it. Please find our responses below.
>
>
>
> <Comment 1> Therefore, the improvement may be caused by the data distribution that is more suitable for geometric ratio, i.e., more younger faces and less older faces. Table 8 also provides such comparison, but please clarify the testing dataset.
> ==> In the balanced dataset, as in FG-NET, there are more faces of younger people. The table below summarizes the distribution. Thus, we agree with the reviewer that the improvement is partly due to the non-uniform distribution of ages.
>
>     ----------------------------------------------------------------------------------------
>     Test data in the balanced dataset
>     ----------------------------------------------------------------------------------------
>     Age range     10s      20s      30s      40s     50s      60s     70s     Total
>     # of faces      790     2,387   2,175   991     667     273     136     7,429
>     ----------------------------------------------------------------------------------------
>
>     ----------------------------------------------------------------------------------------
>     FG-Net database
>     ----------------------------------------------------------------------------------------
>     Age range     0s       10s      20s     30s     40s     50s     60s     Total
>     # of faces      371     339     144     79       46       15       8         1,002
>     ----------------------------------------------------------------------------------------
>
> ==> As mentioned in Appendix D.3, the scores in Table 8 are poorer than those in Table 3 since the comparator is trained for a smaller number of epochs. If we train the arithmetic scheme comparator with sufficiently, it yields MAE = 4.25 and CS=73.0. Thus, we have
>
>     --------------------------------------------------------
>     MV                                MAE=4.49    CS=69.9
>     Geometric scheme    MAE=4.23    CS=73.2
>     Arithmetic scheme    MAE=4.25    CS=73.0
>     --------------------------------------------------------
>
> The arithmetic scheme also outperforms MV, which indicates that the improvement is not just because of the geometric ratio. This will be clarified in the camera-ready.
>
>
>
> <Comment 2> In Algorithm 1, are you using a pre-trained comparator? ... implies a relatively small mutual shift. However, if encounter a large shift, ... will the proposed unsupervised learning converge?
> ==> No, we do not use a pre-trained comparator. As the reviewer pointed out, there is a relatively small shift between chains in age estimation.
> ==> If there is a larger shift, it means that chains are more clearly discernible from one another. In such a case, we expect the unsupervised algorithm would separate clusters (chains) more easily. However, we need to find such examples and do more investigation.
>
>
>
> <Comment 3> It would be great if the authors could provide more insight to the 2CH unsupervised results. How the two chains are divided by analyzing the trained model? What is the key to divide the data? Maybe, it is like a black-box that is hard to explain but this will not affect the merit of this paper.
> ==> As mentioned in the 2nd paragraph on page 10, in the unsupervised 2CH result with $\kappa=10\%$, `looking-older' people are separated from `looking-younger' people. We have not found such clear interpretation for $\kappa=50\%$ yet, but conjecture that apparent ages affect the chain membership in this case as well. We are currently analyzing the experimental results to verify this conjecture.
>
>
>
> Thank you again for your constructive comments.

---

### Decision · Program_Chairs · 2019-12-19

**Decision:**

Accept (Poster)

**Comment:**

This paper addresses a promising method for order learning and applies the new ideas of multiple-chain learning and anchor selection to age estimation and aesthetic regression. The decision regarding instance class is made by comparing it with anchor instances in the same chain and maximizing the consistency among the comparison results. In a multi-chain setting, each chain may correspond to a higher-level attribute class, for example, gender or ethnic group. Supervised and unsupervised learning of multiple ordered chains is proposed.  As rightly acknowledged by R4: “What more promising is the unsupervised chains, which could automatically search for a more optimal multi-chain division scheme than the pre-defined data division.”
All three reviewers and AC agree that the proposed approach is interesting and shows promising results. There are several potential weaknesses and suggestions to further strengthen this work:
(1) more quantitative results are needed for assessing the benefits of this approach (R3, R4) -- see R3’s request to complete the results for FG-Net, to include the results for CLAP2016 and a comparison with the SOTA method BridgeNet. Pleased to report that the authors have revised the manuscript and have included performance of the arithmetic scheme as well as the geometric scheme for FG-Net. Also the authors have provided some initial evaluations of BridgeNet and promised to report the final results as well as the results for CLAP2016 in the final version.
(2) R3 and R4 have expressed concerns regarding using the geometric ratio of the class distances in age estimation and that the improvement may be caused by the data distribution that favours it (R4) or because the baseline methods are not fine-tuned in the same manner (R3). The authors have partially addressed this concern in the rebuttal.
There is a large body of work in computer vision that is focused on relative comparison of samples based on attributes (e.g. age) that is not clearly articulated in the discussions / baseline comparisons (1CH) -- see the seminal work [Relative attributes by Parikh and Grauman, ICCV2011] and the follow up works.
Considering the author response, the AC decided that the most crucial concerns have been addressed in the revision and that the paper could be accepted, but the authors are strongly urged to include additional results that were promised in the rebuttal for the final revision.